# Fast wide-field upconversion luminescence lifetime thermometry enabled by single-shot compressed ultrahigh-speed imaging

Xianglei Liu[1,3], Artiom Skripka [1,2,3], Yingming Lai[1], Cheng Jiang[1], Jingdan Liu [1], Fiorenzo Vetrone[1✉] &
Jinyang Liang[1✉]

Photoluminescence lifetime imaging of upconverting nanoparticles is increasingly featured in recent progress in optical thermometry. Despite remarkable advances in photoluminescent temperature indicators, existing optical instruments lack the ability of wide-field photo-luminescence lifetime imaging in real time, thus falling short in dynamic temperature mapping. Here, we report video-rate upconversion temperature sensing in wide field using single-shot photoluminescence lifetime imaging thermometry (SPLIT). Developed from a compressed-sensing ultrahigh-speed imaging paradigm, SPLIT first records wide-field lumi-nescence intensity decay compressively in two views in a single exposure. Then, an algo-rithm, built upon the plug-and-play alternating direction method of multipliers, is used to reconstruct the video, from which the extracted lifetime distribution is converted to a tem-perature map. Using the core/shell $NaGdF_4$:$Er^{3+}$,$Yb^{3+}$/$NaGdF_4$ upconverting nanoparticles as the lifetime-based temperature indicators, we apply SPLIT in longitudinal wide-field temperature monitoring beneath a thin scattering medium. SPLIT also enables video-rate temperature mapping of a moving biological sample at single-cell resolution.

[1] Centre Énergie Matériaux Télécommunications, Institut National de la Recherche Scientifique, 1650 boulevard Lionel-Boulet, Varennes, Québec J3X1S2, Canada. [2] Present address: Nanomaterials for Bioimaging Group, Departamento de Física de Materiales, Facultad de Ciencias, Universidad Autónoma de Madrid, Madrid, 28049, Spain and The Molecular Foundry, Lawrence Berkeley National Laboratory, Berkeley, CA 94720, USA. [3] These authors contributed equally: Xianglei Liu, Artiom Skripka. ✉email: Fiorenzo.Vetrone@inrs.ca; Jinyang.Liang@inrs.ca

Temperature is an important parameter associated with many physical, chemical, and biological processes[1]. Accurate and real-time (i.e., the actual time during which the event occurs) temperature sensing at microscopic scales is essential to both industrial applications and scientific research, including the examination of internal strains in turbine blades[2], control of the synthesis of ionic liquids[3], and theranostics of cancer[4]. In the past decade, photoluminescence lifetime imaging (PLI) has emerged as a promising approach to temperature sensing[5]. Because photoluminescence can be both excited and detected optically, the resulting non-contact PLI possesses a high spatial resolution[6–8]. This advantage not only overcomes the intrinsic limitation in spatial resolution of imaging thermography due to the long wavelengths of thermal radiation but also avoids heat-transfer-induced inaccuracy in conventional contact methods[9]. Moreover, independent of prior knowledge of samples' physical properties (e.g., emissivity and Grüneisen coefficient[10,11]), PLI brings in higher flexibility in sample selection. Furthermore, PLI is less susceptible than the intensity-based measurements to inhomogeneous signal attenuation, stray light, photobleaching, light's path length, and excitation intensity variations[12–16]. Finally, PLI does not rely on the concentration of labeling agents[8], which eliminates the need for special ratiometric probes[17]. Overcoming many challenges in previous methods, PLI is becoming a popular choice for optical thermometry[18–22].

The success of PLI in temperature mapping depends on two essential constituents: temperature indicators and optical imaging instruments. Recent advances in biochemistry, materials science, and molecular biology have discovered numerous labeling agents[23–26] for PLI-based temperature sensing. Among them, lanthanide-doped upconverting nanoparticles (UCNPs) are ideal candidates. Leveraging the long-lived excited states provided by the lanthanide ions, UCNPs can sequentially absorb two (or more) low-energy near-infrared photons and convert them to one higher-energy photon. This upconversion process allows using excitation power densities several orders of magnitude lower than those needed for simultaneous multi-photon absorption[27,28]. The near-infrared excitation, with smaller tissue extinction coefficients, also gains deeper penetration[29]. Besides, the upconverted luminescence, particularly the Boltzmann-coupled emission bands in co-doped erbium/ytterbium ($Er^{3+}$/$Yb^{3+}$) systems, is highly sensitive to temperature changes[30,31]. Moreover, long-lived (i.e., microseconds to milliseconds) photoluminescence of UCNPs circumvents interferences from autofluorescence and scattering during image acquisition, which translates into improved imaging contrast and detection sensitivity. Finally, because of advances in their synthesis and surface functionalization coupled with the innovation of core/shell engineering, over the years, UCNPs have become much brighter, photostable, biocompatible, and non-toxic[32]. As a result of these salient merits, UCNPs are one of the frontrunners in temperature indicators for PLI.

Advanced optical imaging is the other indispensable constituent in PLI-based temperature mapping[33]. To detect photoluminescence on the time scale of microseconds to milliseconds, like that produced by UCNPs, most PLI techniques use point-scanning time-correlated single-photon counting (TCSPC)[34]. Although they possess high signal-to-noise ratios, the scanning operation leads to an excessively long imaging time to form a two-dimensional (2D) lifetime map because extended pixel dwell time is required to record the long-lived emission[35]. To accelerate data acquisition, wide-field PLI modalities based on parallel collection in time-domain and frequency-domain have been developed[36]. In the time domain, these techniques extend the TCSPC technique to wide-field imaging (e.g., TimepixCam[37] and Tpx3Cam[38]). Photoluminescence decay over a 2D field of view (FOV) is synthesized from >100,000 frames, which requires the

emission to be precisely repeatable. Alternatively, the frequency-domain wide-field PLI techniques[39,40] use phase difference between the intensity-modulated excitation and the received photoluminescence signal to determine the 2D lifetime distribution. Nevertheless, limited by the range of frequency synthesizers, the measurable lifetimes are mostly restricted to ≤100 µs, which is shorter than the lifetimes of most UCNPs. Akin to the time-domain techniques, these systems rely on the integration over many periods of modulation intensity, during which the sample must remain stationary. Thus far, existing PLI techniques fall short in 2D temperature sensing of moving samples with a micrometer-level spatial resolution.

To surmount these limitations, we report an optical temperature mapping modality, termed single-shot photoluminescence lifetime imaging thermometry (SPLIT). Synergistically combining dual-view optical streak imaging with compressed sensing[41], SPLIT records wide-field luminescence decay of $Er^{3+}$, $Yb^{3+}$ co-doped $NaGdF_4$ UCNPs in real time, from which a lifetime-based 2D temperature map is obtained in a single exposure. Largely advancing existing optical thermometry techniques in detection capabilities, SPLIT enables longitudinal 2D temperature monitoring beneath a thin scattering medium and dynamic temperature tracking of a moving biological sample at single-cell resolution.

## Results

**Operating principle of SPLIT.** The schematic of the SPLIT system is shown in Fig. 1. A 980-nm continuous-wave laser (BWT, DS3-11312-113-LD) is used as the light source. The laser beam passes through a $4f$ system consisting of two 50-mm focal length lenses (L1 and L2, Thorlabs, LA1255). An optical chopper (Scitec Instruments, 300CD) is placed at the back focal plane of lens L1 to generate 50-µs optical pulses. Then, the pulse passes through a 100-mm focal length lens (L3, Thorlabs, AC254-100-B) and is reflected by a short-pass dichroic mirror (Edmund Optics, 69-219) to generate a focus on the back focal plane of an objective lens (Nikon, CF Achro 4×, 0.1 numerical aperture, 11-mm field number). This illumination scheme produces wide-field illumination ($1.5 \times 1.5$ mm$^2$ FOV) to UCNPs at the object plane.

The near-infrared excited UCNPs emit light in the visible spectral range. The decay of light intensity over the 2D FOV is a dynamic scene, denoted by $I(x, y, t)$. The emitted light is collected by the same objective lens, transmits through the dichroic mirror, and is filtered by a band-pass filter (Thorlabs, MF542-20 or Semrock, FF01-660/30-25). Then, a beam splitter (Thorlabs, BS013) equally divides the light into two components. The reflected component is imaged by a CMOS camera (FLIR, GS3-U3-23S6M-C) with a camera lens (Fujinon, HF75SA1) via spatiotemporal integration (denoted as the operator **T**) as View 1, whose optical energy distribution is denoted by $E_1(x_1, y_1)$.

The transmitted component forms an image of the dynamic scene on a transmissive encoding mask with a pseudo-random binary pattern (Fineline Imaging, 50% transmission ratio; 60-µm encoding pixel size). This process of spatial encoding is denoted by the operator **C**. Then, the spatially encoded scene is imaged by a mechanical streak camera. In particular, the scene is relayed to the sensor plane of an electron-multiplying (EM) CCD camera (Nüvü Camēras, HNü 1024) by a $4f$ imaging system consisting of two 100-mm focal length lenses (L4 and L5, Thorlabs, AC254-100-A). A galvanometer scanner (Cambridge Technology, 6220H), placed at the Fourier plane of the $4f$ imaging system, temporally shears the spatially encoded frames linearly to different spatial locations along the $x_2$ axis of the EMCCD camera according to their time of arrival. This process of temporal shearing is denoted by the operator **S**. Finally, the spatially encoded and temporally sheared dynamic scene is

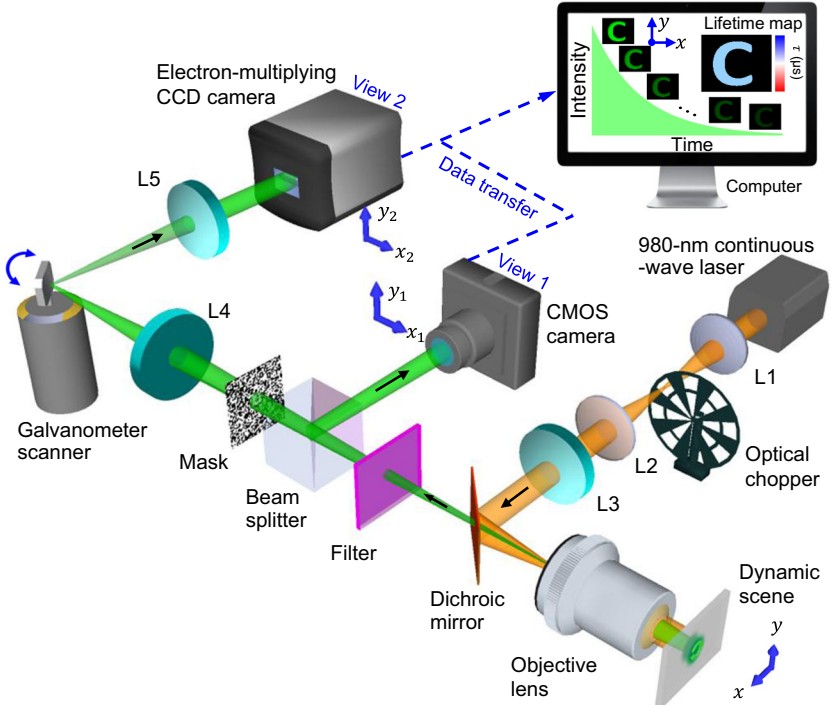

**Fig. 1 Schematic of the SPLIT system.** The illustration shows data acquisition and image reconstruction of luminescence intensity decay in a letter "C". L1–L5, lens.

recorded by the EMCCD camera via spatiotemporal integration to form View 2, whose optical energy distribution is denoted by $E_2(x_2, y_2)$.

By combining the image formation of $E_1(x_1, y_1)$ and $E_2(x_2, y_2)$, the data acquisition of SPLIT is expressed by

$$E = \mathbf{TM}I, \qquad (1)$$

where $E$ denotes the concatenation of measurements $[E_1, \alpha E_2]^T$ (the superscript $T$ denotes the transpose), $\mathbf{M}$ denotes the linear operator $[\mathbf{1}, \alpha \mathbf{SC}]^T$, and $\alpha$ is a scalar factor introduced to balance the energy ratio between the two views during measurement[42]. The hardware of the SPLIT system is synchronized for capturing both views (detailed in "Methods") that are calibrated before data acquisition (detailed in Supplementary Note 1 and Supplementary Fig. 1).

After data acquisition, $E$ is processed by an algorithm that retrieves the datacube of the dynamic scene by leveraging the spatiotemporal sparsity of the dynamic scene and the prior knowledge of each operator[43,44]. Developed from the plug-and-play alternating direction method of multipliers (PnP-ADMM) framework[45,46], the reconstruction algorithm of SPLIT solves the minimization problem of

$$\hat{I} = \underset{I}{\arg\min}\left\{\frac{1}{2}\|\mathbf{TM}I - E\|_2^2 + R(I) + \mathbf{I}_+(I)\right\}. \qquad (2)$$

Here, $\|\cdot\|_2$ represents the $l_2$ norm. The fidelity term, $\frac{1}{2}\|\mathbf{TM}I - E\|_2^2$, represents the similarity between the measurement and the estimated result. $R(\bullet)$ is the implicit regularizer that promotes sparsity in the dynamic scene. $\mathbf{I}_+(\cdot)$ represents a non-negative intensity constraint. Compared to existing reconstruction schemes[47–49], PnP-ADMM implements a variable splitting strategy with a state-of-the-art denoiser to obtain fast and closed-form solutions to each sub-optimization problem, which produces a high image quality in reconstruction (see Supplementary Notes 2 and 3 and Supplementary Fig. 2). The retrieved datacube of the dynamic scene has a sequence depth (i.e., the

number of frames in a reconstructed movie) of 12–100 frames, each containing $460 \times 460$ $(x, y)$ pixels. The imaging speed is tunable from 4 to 33 thousand frames per second (kfps) (detailed in "Methods").

The reconstructed datacube is then converted to a photoluminescence lifetime map. In particular, for each $(x, y)$ point, the area under the normalized intensity decay curve is integrated to report the value of the photoluminescence lifetime[50]. Finally, using the approximately linear relationship between the UCNPs' lifetime and the physiologically relevant temperature range (20–46 °C in this work)[51,52], the 2D temperature distribution, $T(x, y)$, is calculated by

$$T(x, y) = c_t + \frac{1}{S_a}\int \frac{\hat{I}(x, y, t)}{\hat{I}(x, y, 0)}\, \mathrm{d}t, \qquad (3)$$

where $c_t$ is a constant, and $S_a$ is the absolute temperature sensitivity[33]. The derivation of Eq. (3) is detailed in Supplementary Note 4. Leveraging the intrinsic frame rate of the EMCCD camera, the SPLIT system can generate lifetime-determined temperature maps at a video rate of 20 Hz.

**Quantification of the system's performance of SPLIT.** We prepared a series of core/shell UCNP samples to showcase SPLIT's capabilities. These UCNPs shared the same $NaGdF_4$: 2 mol% $Er^{3+}$, 20 mol% $Yb^{3+}$ active core of 14.6 nm in size, while differed by the thickness of their undoped $NaGdF_4$ passive shell of 1.9, 3.5, and 5.6 nm (Fig. 2a and detailed in Supplementary Note 5). All of the UCNP samples were of pure hexagonal crystal phase (Supplementary Fig. 3). Under the 980-nm excitation, upconversion emission bands of all samples were measured at around 525/545 nm and 660 nm, which correspond to the $^2H_{11/2}/^4S_{3/2} \rightarrow {}^4I_{15/2}$ and $^4F_{9/2} \rightarrow {}^4I_{15/2}$ radiative transitions, respectively (Fig. 2b–c).

To characterize SPLIT's spatial resolution, we covered the 5.6 nm-thick-shell UCNP sample with a negative USAF resolution target (Edmund Optics, 55-622). Operating at 33 kfps, SPLIT recorded the photoluminescence decay (Supplementary Movie 1).

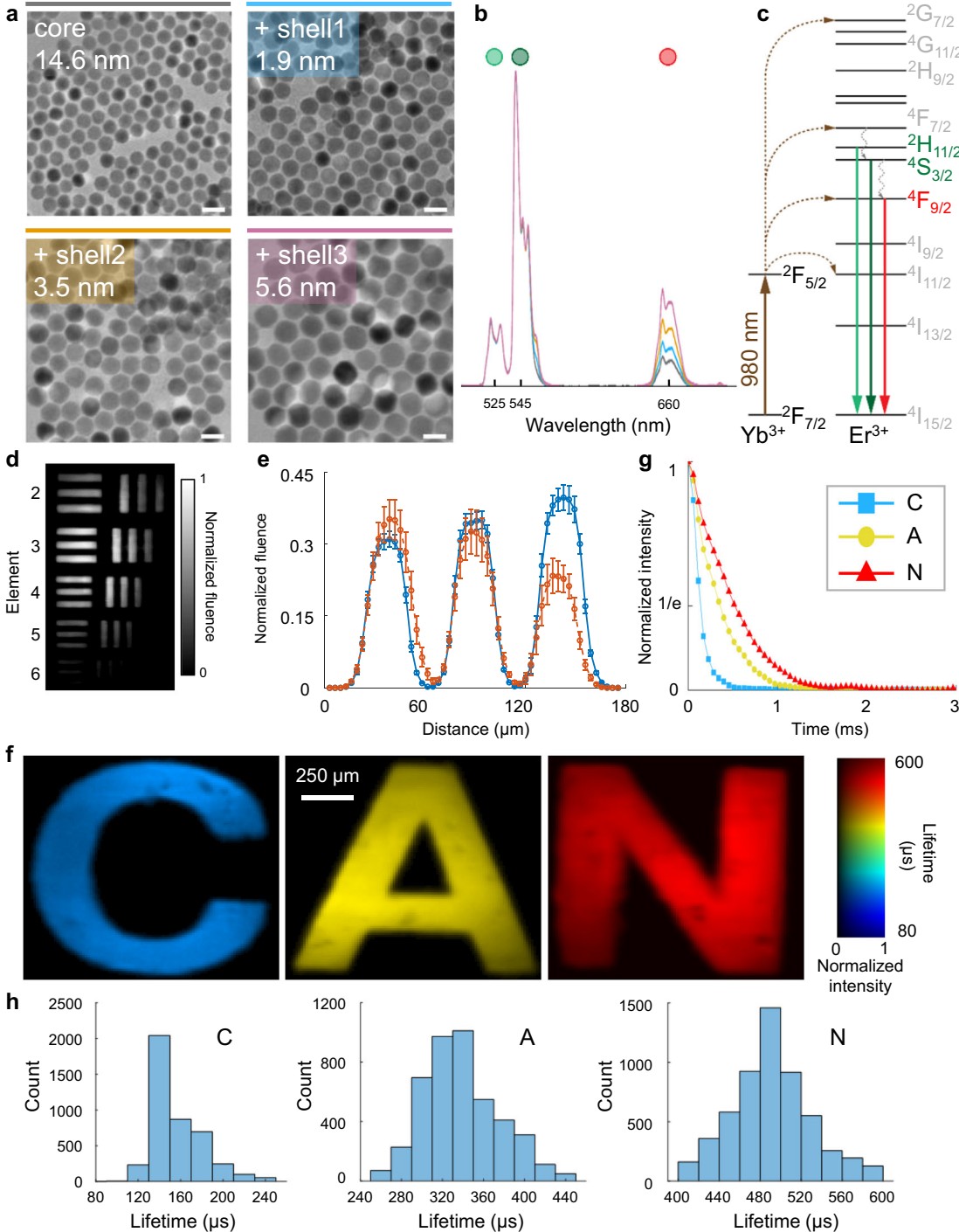

**Fig. 2 Quantification of the performance of the SPLIT system. a** Images of core/shell UCNPs acquired with a transmission electron microscope. Scale bar: 25 nm. **b** Normalized upconversion spectra of UCNPs shown in (**a**). **c** Simplified energy level diagram of $Yb^{3+}$-$Er^{3+}$ energy transfer upconversion excitation and emission. **d** Temporally projected image of photoluminescence intensity decay of the 5.6 nm-thick-shell UCNPs covered by a negative resolution target. **e** Comparison of averaged light fluence distribution along the horizontal bars (blue) and vertical bars (orange) of Element 5 in Group 4 on the resolution target. Error bar: standard deviation. **f** Lifetime images of UCNPs with the shell thicknesses of 1.9, 3.5, and 5.6 nm covered by transparencies of letters "C", "A", and "N" in green emission. **g** Time-lapse averaged emission intensities of the samples. **h** Histograms of photoluminescence lifetimes in the letters shown in (**f**).

The temporally projected datacube reveals that the intensity and contrast in the reconstructed image degrade with the decreased spatial feature sizes, eventually leading to the loss of structure whose size approaches that of the encoding pixel (Fig. 2d). The effective spatial resolution was thus determined to be 20 μm (Fig. 2e). Under these experimental conditions, the minimum

power density for the SPLIT system was quantified to be 0.06 W mm$^{-2}$ (detailed in Supplementary Note 6 and Supplementary Fig. 4).

To demonstrate SPLIT's ability to distinguish different lifetimes, we imaged the UCNPs with shell thicknesses of 1.9, 3.5, and 5.6 nm, covered by transparencies of letters "C", "A", and

"N", respectively, using a single laser pulse (Supplementary Movie 2). The lifetime maps of these samples are shown in Fig. 2f, which reveals the averaged lifetimes for the $^4S_{3/2}$ excited state of samples "C", "A", and "N" to be 142, 335, and 478 μs, respectively (Fig. 2g–h). These results were verified by using the standard TCSPC method (detailed in Supplementary Note 7 and Supplementary Fig. 5).

SPLIT's reconstruction algorithm shows a superb performance to existing mainstream algorithms popularly used in single-shot compressed ultrafast imaging[41,42,47,48]. By using the experimental data, the comparison demonstrates that the dual-view PnP-ADMM algorithm used by SPLIT is more powerful in preserving spatial features while maintaining a low background, which enables a more accurate lifetime quantification and the ensuing temperature mapping (detailed in Supplementary Note 8 and Supplementary Fig. 6).

**Single-shot temperature mapping using SPLIT**. We used the 5.6 nm-thick-shell UCNPs as the temperature indicator for SPLIT. The UCNPs' temperature was controlled by a heating plate placed behind the sample. To image the green ($^4S_{3/2}$) and red ($^4F_{9/2}$) upconversion emissions, the sample was covered by transparencies of a lily flower and a maple leaf, respectively. The temperature of the entire sample was measured with both a Type K thermocouple (Omega, HH306A) and a thermal camera (FLIR, E4) as references. The reconstructed lifetime images in the 20–46 °C temperature range are shown in Fig. 3a–b (see the full evolution in Supplementary Movie 3). Plotted in Fig. 3c–d, the time-lapse averaged intensity over the entire FOV shows that the averaged lifetimes of green and red emissions decrease from 489 to 440 μs and from 458 to 398 μs, which is due to their enhanced multi-phonon deactivation at higher temperatures. We further plotted the relationship between the temperatures and lifetimes for both emission channels (Fig. 3e). Finally, the temperature sensitivities in the preset temperature range were calculated to be $S_a = -1.90\,\mu s\,°C^{-1}$ for the green emission and $S_a = -2.40\,\mu s\,°C^{-1}$ for the red emission (see detailed calculation and further analysis in Supplementary Note 9 and Supplementary Fig. 7). Compared to the green emission, the higher temperature sensitivity of the red emission results from the greater energy separation between its emitting state and the adjacent lower-laying excited state (Fig. 2c). Since multi-phonon relaxation rate depends exponentially on the number of phonons necessary to deactivate an excited state to the one below it, the increase in phonon energies at higher temperatures has greater influence over the states with a larger energy gap between them[53]. These results establish lifetime-temperature calibration curves [i.e., Eq. (3)] for ensuing thermometry experiments.

To demonstrate SPLIT's feasibility in a biological environment, we conducted longitudinal temperature monitoring under a phantom, made by using the 5.6 nm-thick-shell UCNPs covered by lift-out grids (Ted Pella, 460-2031-S), overlaid by fresh chicken breast tissue. We investigated SPLIT's imaging depth with varied tissue thicknesses of up to 1 mm (Fig. 3f, Supplementary Note 10, Supplementary Fig. 8, and Supplementary Movie 4). The chicken tissue of 0.5 mm thickness, where both the green and red emissions produced images with full spatial features of the lift-out grid, was used in the following imaging experiments. Subsequently, we cycled the temperature of the sample between 20 and 46 °C. The lifetime distributions of both green and red emissions and their corresponding temperature maps were monitored every 20 and 23 min, respectively, for ~4 h (see the full evolution in Supplementary Fig. 9 and Supplementary Movie 5). As shown in Fig. 3g, the results are in good agreement with the temperature change preset by the heating plate, and decisively showcase how SPLIT can map 2D temperatures over time with high accuracy beneath biological tissue.

We also demonstrated SPLIT using a fresh beef phantom as a scattering medium, where both light scattering and absorption are present (detailed in Supplementary Note 10 and Supplementary Fig. 10). The results reveal better penetration of the red emission over the green counterpart due to its weaker scattering and absorption. More importantly, the results confirm the independence of the measured photoluminescence lifetime of UCNPs to tissue thickness and hence the excitation light power density used in our work (≤0.4 W mm$^{-2}$).

**Single-cell dynamic temperature tracking using SPLIT**. To apply SPLIT to dynamic single-cell temperature mapping, we tested a single-layer onion epidermis sample labeled by the 5.6 nm-thick-shell UCNPs (detailed in Supplementary Note 11 and Supplementary Fig. 11). Furthermore, to generate non-repeatable photoluminescent dynamics, the sample was moved across the FOV at a speed of 1.18 mm s$^{-1}$ by a translation stage. In the 3-second measurement window, the SPLIT system continuously recorded 60 lifetime/temperature maps. Four representative time-integrated images and their corresponding lifetime maps are shown in Fig. 4a–b (see dynamic lifetime mapping in Supplementary Movie 6). Figure 4c shows intensity decay curves from four selected regions with varied intensities in the onion cell sample at 0.05 s. The photoluminescence lifetimes and hence the temperatures remain stable, showing SPLIT's resilience to spatial intensity variation. We also tracked the time histories of the average emitted fluence and lifetime-indicated temperatures of these four regions during the sample's translational moving (Fig. 4d). In this measurement time window, the emitted photoluminescence fluences have varied in each selected region. In contrast, the measured temperatures show a small fluctuation of ±0.35 °C, which validates the advantage of PLI thermometry in handling temporal intensity variation.

## Discussion

In summary, we have developed SPLIT for wide-field dynamic temperature sensing in real time. In data acquisition, SPLIT compressively records the photoluminescence emission over a 2D FOV in two views. Then, the dual-view PnP-ADMM algorithm reconstructs spatially resolved intensity decay traces, from which a photoluminescence lifetime distribution and the corresponding temperature map are extracted. Used with core/shell NaGdF$_4$:Er$^{3+}$,Yb$^{3+}$/NaGdF$_4$ UCNPs, SPLIT has enabled temperature mapping with high sensitivity for both green and red upconversion emission bands with a 20-μm spatial resolution in a 1.5 × 1.5 mm$^2$ FOV at a video rate of 20 Hz. SPLIT is demonstrated in longitudinal temperature monitoring of a phantom beneath fresh chicken tissue. SPLIT is also applied to dynamic single-cell temperature mapping of a moving single-layer onion epidermis sample.

SPLIT advances the technical frontier of optical instrumentation in PLI. The high parallelism in SPLIT's data acquisition drastically improves the overall light throughput. The resulting system, featuring single-shot temperature sensing over a 2D FOV, solves the long-standing issue in scanning-based techniques (see Supplementary Note 12 and Supplementary Figs. 12–13). In particular, SPLIT improves the measurement accuracy by avoiding artifacts generated from the scanning-induced motion blur and the excitation intensity fluctuation. More importantly, as shown in Fig. 4, SPLIT extends the application scope of PLI to observing non-repeatable 2D temperature dynamics. Its high tunability of imaging speeds also accommodates a variety of UCNPs with a wide lifetime span (from hundreds of nanoseconds to milliseconds). Among existing single-shot 2D ultrafast imaging

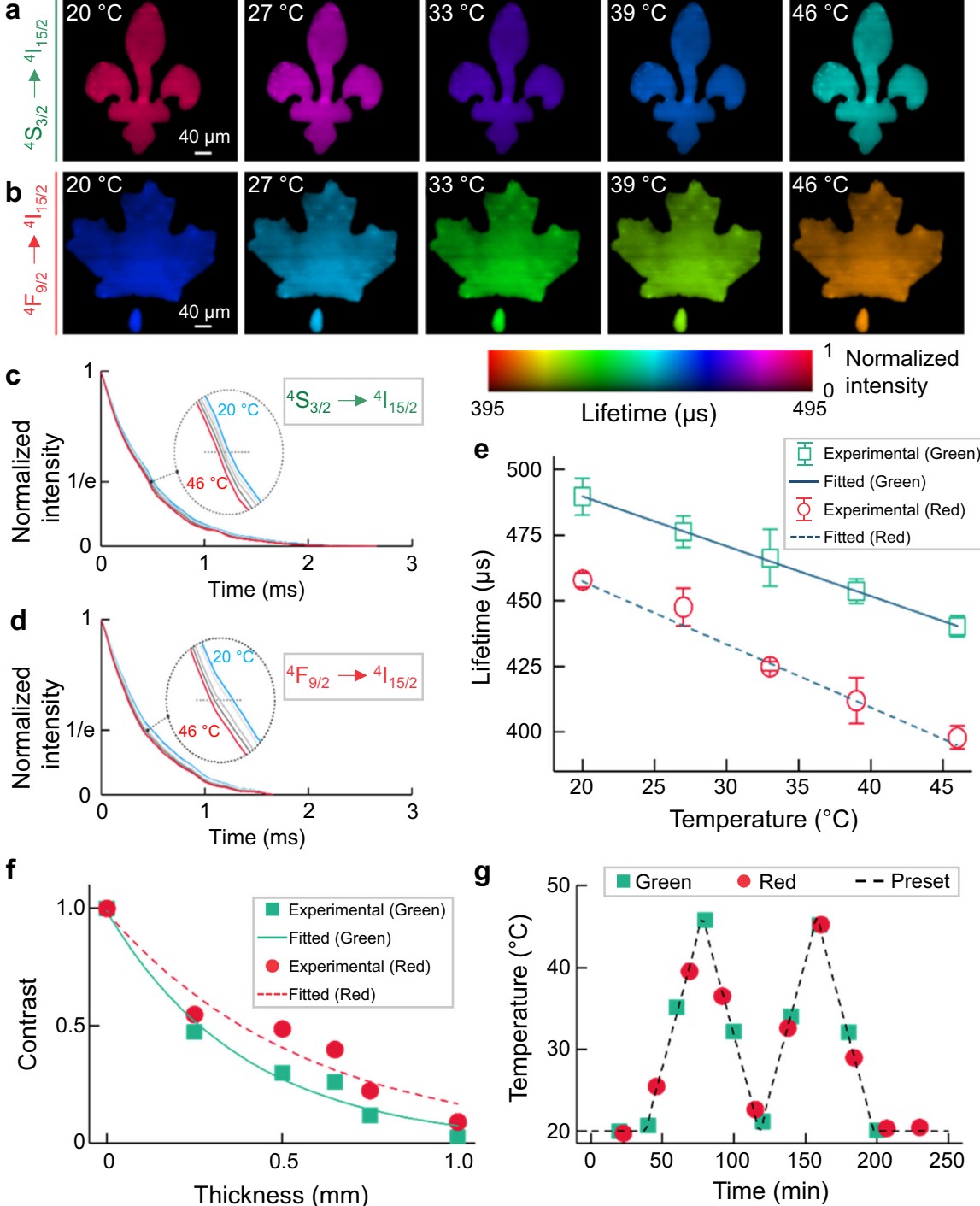

**Fig. 3 Single-shot temperature mapping using SPLIT. a, b** Lifetime images of green (**a**) and red (**b**) upconversion emission bands under different temperatures. **c, d** Normalized photoluminescence decay curves of green (**c**) and red (**d**) emission bands at different temperatures, averaged over the entire field of view. **e** Relationship between temperature and mean lifetimes of green and red emissions with linear fitting. Error bar: standard deviation from three independent measurements. **f** Normalized contrast versus chicken tissue thickness for green and red emission bands with single-component exponential fitting. **g** Longitudinal temperature monitoring through 0.5 mm-thick fresh chicken tissue.

modalities based on streak cameras, SPLIT is well suited for dynamic PLI of UCNPs in terms of the targeted imaging speed, detection sensitivity, spatial resolution, and cost efficiency (detailed in Supplementary Note 12 and Supplementary Table 1). Finally, the SPLIT system by itself records only the lifetime images; yet, when using UCNPs as contrast agents, those images carry temperature information in situ, where the UCNPs reside. Thus, compared to thermal imaging cameras, SPLIT supplies

superior temperature mapping results with higher image contrast and better resilience to background interference (detailed in Supplementary Note 13 and Supplementary Fig. 14).

From the perspective of system design, both the dual-view data acquisition and the PnP-ADMM algorithm support high imaging quality in SPLIT. In particular, View 1 preserves the spatial information in the dynamic scene[54]. Meanwhile, View 2 retains temporal information by optical streaking via time-to-space conversion.

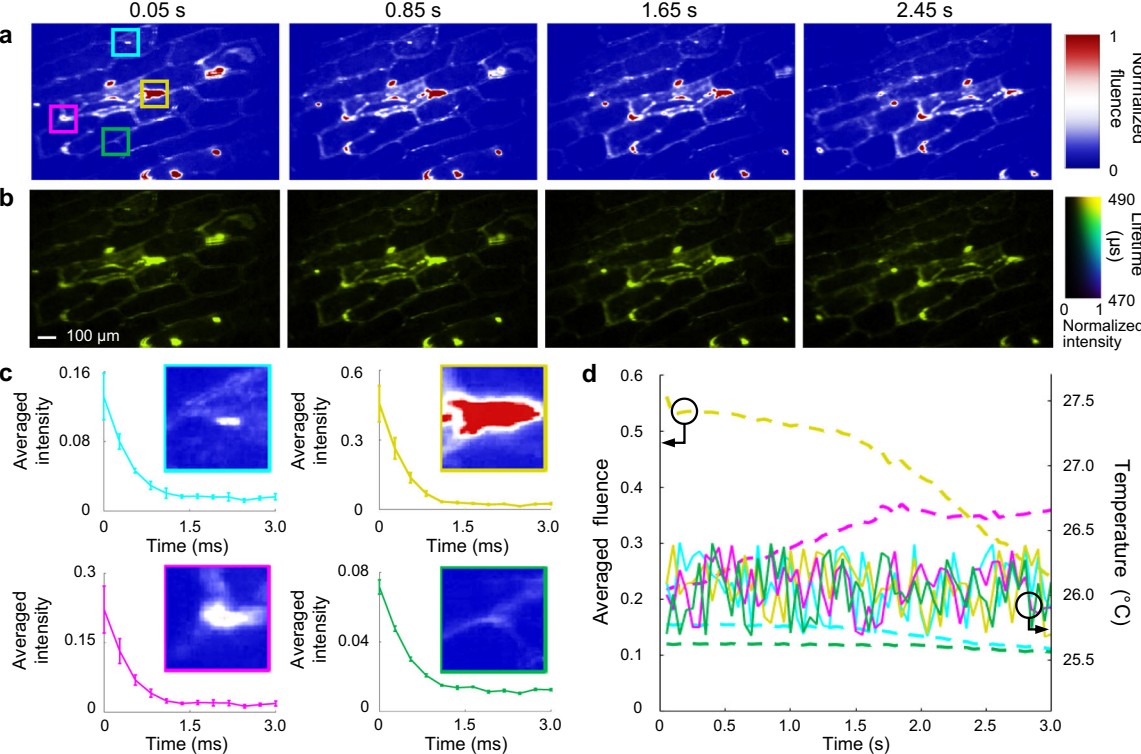

**Fig. 4 Dynamic single-cell temperature mapping using SPLIT. a** Representative time-integrated images of a moving onion epidermis cell sample labeled by UCNPs. **b** Lifetime images corresponding to (**a**). **c** Photoluminescence decay profiles at four selected areas [marked by the solid boxes in the first panel of (**a**)] with varied intensities. **d** Time histories of averaged fluence and corresponding temperature in the four selected regions during the sample's translational motion.

Altogether, both views maximally keep rich spatiotemporal information. In software, the dual-view PnP-ADMM algorithm provides a powerful modular structure, which allows separated optimization of individual sub-optimization problems with an advanced denoising algorithm to generate high-quality image restoration results.

SPLIT offers a versatile PLI temperature-sensing platform. In materials characterization, it could be used in the stress analysis of metal fatigue in turbine blades[55]. In biomedicine, it could be implemented for accurate sub-cutaneous temperature monitoring for theranostics of skin diseases (e.g., micro-melanoma)[56,57]. SPLIT's microscopic temperature mapping ability could also be exploited for the studies of temperature-regulated cellular signaling[58]. Finally, the operation of SPLIT could be extended to Stokes emission in lanthanide-doped nanoparticles and spectrally resolved temperature mapping. All of these topics are promising research directions in the future.

## Methods

**Synchronization of the SPLIT system**. The optical chopper outputs a transistor-transistor logic (TTL) signal that is synchronized with the generated optical pulses. This TTL signal is input to a delay generator (Stanford Research Systems, DG 645), which then generates three synchronized TTL signals at 20 Hz. The first two signals are used to trigger the 3-ms exposure of the EMCCD and CMOS cameras. The last one is used to trigger a function generator (Rigol, DG1022Z) that outputs a 20-Hz sinusoidal waveform under the external burst mode to control the rotation of the galvanometer scanner (GS).

**Calculation of SPLIT's key parameters**. The GS, placed at the Fourier plane of the 4*f* imaging system consisting of lenses L4 and L5 (Fig. 1), deflects temporal information to different spatial positions. Rotating during the data acquisition, the GS changes the reflection angles of the spatial frequency spectra of individual frames with different time-of-arrival. After the Fourier transformation by Lens 5, this angular difference is converted to the lateral shift in space on the EMCCD camera, which results in temporal shearing. An illustration with a simple example is provided in Supplementary Fig. 15.

The imaging speed is determined by the data acquisition for View 2. In particular, the reconstructed movie has a frame rate of[59]

$$r = \frac{\gamma_a V_g f_5}{t_s d}. \tag{4}$$

Here $V_g$ is the voltage added onto the GS. $\gamma_a$ is a constant that links $V_g$ with GS's deflection angle with the consideration of the input waveform. $f_5 = 100$ mm is the focal length of lens L5, $t_s = 50$ ms is the period of the sinusoidal voltage waveform added to the GS, and $d = 13\,\mu m$ is the EMCCD sensor's pixel size. In this work, we used the voltage from $V_g = 0.2–1.7$ V. The imaging speed of SPLIT ranged from 4 to 33 kfps. In addition, we used $t_e = 3$ ms as the exposure time of the EMCCD and CMOS cameras. The sequence depth, $N_t$, is determined by

$$N_t = rt_e. \tag{5}$$

In the experiments presented in this work, $N_t$ ranged from 12 to 100 frames.

## Data availability

All data needed to evaluate the findings of this study are present in the paper and Supplementary Information. The raw data for Fig. 2 can be downloaded via the following link: https://figshare.com/articles/figure/SPLIT_Fig2/16703413. All other raw data are available from the corresponding authors upon reasonable request.

## Code availability

The image reconstruction algorithm is described in detail in Supplementary Information. The custom computer code is not publicly available because it is proprietary and included in a patent application.

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

## Acknowledgements

The authors thank Professor Aycan Yurtsever and Wanting He for experimental assistance and fruitful discussion. Natural Sciences and Engineering Research Council of Canada (RGPIN-2017-05959, RGPAS-2017-507845, I2IPJ-555593-20, RGPIN-2018-06217, RGPAS-2018-522650); Canada Foundation for Innovation and Ministère de l'Économie et de l'Innovation du Québec (37146); Canadian Cancer Society (707056); New Frontier in Research Fund (NFRFE-2020-00267); Fonds de Recherche du Québec–Nature et Technologies (2019-NC-252960); Fonds de Recherche du Québec–Santé (267406, 280229).

## Author contributions

X.L. designed and built the system, conducted the experiments, developed the reconstruction algorithm, and analyzed the data. A.S. prepared the UCNPs, conducted some experiments, and analyzed the data. Y.L. contributed to the algorithm development. C.J. and J. Liu conducted some experiments. F.V. and J. Liang initiated the project. J. Liang proposed the concept, contributed to experimental design, and supervised the project. All authors wrote and revised the manuscript.

## Competing interests

The authors disclose the following patent applications: WO 2020/154806 A1 (J. Liang, F.V., and X.L.) and US Provisional 63/260,511 (J. Liang, F.V., X.L., and A.S.).
