## [Peer Review File · Nature Communications]

REVIEWER COMMENTS

Reviewer #1 (Remarks to the Author):

The authors present a novel compressed sensing (CS) imaging technique capable of sensing phosphorescence lifetime (PLT) with ~ 30 microseconds temporal resolution (i.e. 33 kfps). This imaging technique captures PLT from an area of 1.5mm by 1.5mm with a spatial resolution of 20 microns. Optical characterization of the system is presented by imaging upconverting nanoparticles (UCNP) in both biological samples and in engineering samples, demonstrating its accuracy and precision of the complete imaging system.

This imaging technique is an extension of the one described in references [41] and [42] with important improvements: the expensive streak element in the original work is replaced with an inexpensive galvanometer. This replacement has enabled the authors to create much more compact and affordable imaging system at the cost of lower temporal resolution (100 psec temporal resolution vs. 30 microseconds). This is an important contribution which can enable researching phenomena at higher temporal framerate compare to sCMOS camera and at lower cost.

There are several concerns that the authors should clarify:

1. The difference between the work in [41], [42] or prior work on ultrafast imaging using streak camera should be clearly articulated. What are the benefits and tradeoffs of your work compared to the streak approach? Among other factors, temporal speed, spatial resolution, size, power and cost should be considered.
2. What is the main difference in the software reconstruction algorithm in this work compared to the ones used in [41], [42] or prior work on ultrafast imaging using streak cameras. A state-of-the-art denoising algorithm is used in this work. This claim should be quantified and experimental data should be provided.
3. High resolution imaging is claimed in both the abstract and throughout the paper. This has to be put in the context of an imaged area, number of pixels in the imaging systems, optics, etc.
4. The authors make a claim that this image sensor can be used for high accuracy thermal imaging and cite several papers where thermal imaging is critical. However, the paper provides convincing evidence that the imaging system can detect thermally modulated PLT. In other words, a UCNPs are necessary to sense the temperature of the underlying structure under the assumption that both entities have the same temperature. This is different when compared to how thermal cameras operate. These claims are misleading and should be corrected.
5. The key feature of this imaging technology is the PLT imaging capabilities with ~ 30 microseconds temporal resolution and leveraging sparsity in a scene to employ CS algorithms. This is the key strength of the work and achieved with an elegant approach. It would be good for this to be better explained rather than the focus to be on imaging temperature modulated PLT. The temperature modulated PLT can be part of the presented work but not necessarily the focus.
6. The time shearing effects should be better explained in the main text or methods. Since temporal and spatial resolution are tightly coupled in this imaging system, they should be elaborated in the text. It is hard to understand how was 33 kfps was achieved: the EMCCD operates at 16 fps and has 1024 by 1024 pixels. If the spatial resolution is 20 microns on a 1500 by 1500 micron sample, then the effective number of pixels are 75 by 75. This means temporal resolution can only be increased by a factor of 186 by time shearing or ~ 3 kfps will be the maximum frame rate. How was achieved higher temporal resolution? What is the effective spatial resolution?
7. The experiments of imaging UCNPs under different chicken phantoms should be improved. A better scattering medium should be chosen since the chosen phantoms don't have water or blood which are the main scattering culprits.
8. The proposed imaging system should be more carefully evaluated against high-speed commercial cameras (ref. 36, 37 and other cameras with fps of ~ 100 K). Also if the argument about temperature sensing in PLT is maintained in the paper, then a comparison with FLIR camera should be provided in the paper (spatial and temporal resolution, sensitivity, size, power, cost, etc).

Reviewer #2 (Remarks to the Author):

Liu et al. reported a time-domain wide-field lifetime imaging system. Combined with Yb³⁺/Er³⁺ nanothermometers, this system realized the intracellular temperature mapping in onion cells. This is an interesting work, but this reviewer feels that the originality and experimental findings did not meet the standards of Nature Communications.

1. The mechanical streaking-based lifetime imaging introduced in this work was reported previously with the same principle (Opt. Express 2020, 28, 26717-26723). Instead of using a mask, the previous reported work used a line-scanning confocal set-up.
2. The thermal responsive mechanism of Yb³⁺/Er³⁺ is not studied in this work, however, similar phenomena was reported before (Nature Photon. 2018, 12, 154–158).
3. The thermal uncertainty (or thermal resolution) by lifetime measurement is not given in this context.
4. The upconversion process is power dependent, and the lifetime of the green or red emission band of Er via upconversion changes in different excitation power density. The established lifetime-temperature values can not be applied in different power density (like through the chicken breast).
5. Although this work demonstrated the experiments with chicken breast, as widely acknowledged, the emission wavelength in the visible region is not suitable for deep tissue imaging.

Reviewer #3 (Remarks to the Author):

The authors have developed a new luminescence lifetime imaging technique that combines controlled illumination and smart image acquisition through a coded aperture to enable compressed sampling of image data in space and time at extreme temporal resolution.

The authors should compare their work to this particular prior publication: Lixin Liu, Yahui Li, Luogeng Sun, Heng Li, Xiao Peng, Junle Qu, "Fluorescence lifetime imaging microscopy using a streak camera," Proc. SPIE 8948, Multiphoton Microscopy in the Biomedical Sciences XIV, 89482L (28 February 2014); <https://doi.org/10.1117/12.2039056>, who used galvomirrors only to attempt to obtain similar results, this may help justify the benefits of an indirect imaging method using scanning and a coded aperture.

The authors should also compare their work to ref. 48, in greater details, since this imaging method also used a coded aperture to intermix spatial and temporal information, perhaps taking away some of the novelty claims of the paper. Is the novelty limited to the denoising algorithm? If so, please clearly indicate if the setup in figure 1 has been previously used by others, and if there are minor variations in the design and operation, please explain what those specific contributions are and how this experimental setup differs from techniques previously published.

Besides those two minor comments, the experiments validate the claims of performance quite clearly, the imaging capabilities are exciting and valuable to a broad community, and this reviewer, therefore, believes that the article is suitable for publication in this journal.

RESPONSES TO REFEREES

We sincerely appreciate the valuable comments from the referees, which we have carefully adopted to improve the quality of our manuscript. In this round of revision, we have included detailed descriptions to explain and compare the technical features of our work. We have also conducted many new experiments to directly address the referees' concerns. These amendments are reflected in the new information and clarification presented in Main Text. As for Supplementary Materials, the major changes include the following points:

- We have added **Supplementary Note 8** and **Supplementary Fig. 6** to experimentally demonstrate the superb performance of the image reconstruction method used in SPLIT compared to existing works.
- We have expanded **Supplementary Notes 9–10** and added **Supplementary Fig. 10** to report the thermal uncertainty, to experimentally demonstrate SPLIT with the presence of both light scattering and absorption, and to experimentally verify the invariance of photoluminescence lifetimes to excitation power density.
- We have added **Supplementary Note 12**, **Supplementary Figs. 12–13**, and **Supplementary Table 1** to articulate and experimentally demonstrate the novelty of SPLIT as well as to distinguish it from existing modalities of streak-camera-based photoluminescence lifetime imaging.
- We have added **Supplementary Note 13** and **Supplementary Fig. 14** to experimentally demonstrate the advantages of SPLIT over conventional thermal imaging.
- We have added **Supplementary Fig. 15** to illustrate the temporal shearing in SPLIT's data acquisition.

Point-by-point responses are provided below, and all changes in the revised manuscript are highlighted in red.

REFEREE #1

[Comment 0]

The authors present a novel compressed sensing (CS) imaging technique capable of sensing phosphorescence lifetime (PLT) with ~30 microseconds temporal resolution (i.e. 33 kfps). This imaging technique captures PLT from an area of 1.5mm by 1.5mm with a spatial resolution of 20

microns. Optical characterization of the system is presented by imaging upconverting nanoparticles (UCNP) in both biological samples and in engineering samples, demonstrating its accuracy and precision of the complete imaging system. This imaging technique is an extension of the one described in references [41] and [42] with important improvements: the expensive streak element in the original work is replaced with an inexpensive galvanometer. This replacement has enabled the authors to create much more compact and affordable imaging system at the cost of lower temporal resolution (100 psec temporal resolution vs. 30 microseconds). This is an important contribution which can enable researching phenomena at higher temporal framerate compare to sCMOS camera and at lower cost. There are several concerns that the authors should clarify:

[Response 0]

We appreciate the referee's acknowledgment of the novelty of our method and the advantages of cost-efficiency, compactness, and measurement accuracy in our setup that contributes to high-speed optical imaging.

[Comment 1]

The difference between the work in [41], [42] or prior work on ultrafast imaging using streak camera should be clearly articulated. What are the benefits and tradeoffs of your work compared to the streak approach? Among other factors, temporal speed, spatial resolution, size, power and cost should be considered.

[Response 1]

We have added a detailed explanation to articulate the difference between SPLIT and previous works that used streak cameras (optoelectronic and mechanical types) for photoluminescence lifetime imaging (including the two references mentioned by the referee as well as other representative works). This comparison is described in terms of related concepts, technical specifications (including imaging speed, spatial resolution, size, power, cost, and others), advantages/limitations, and their applications. Please see this new content in Supplementary Note 12, Supplementary Figs. 12–13, and Supplementary Table 1 in the revised manuscript.

[Comment 2]

What is the main difference in the software reconstruction algorithm in this work compared to the ones used in [41], [42] or prior work on ultrafast imaging using streak cameras. A state-of-the-art denoising algorithm is used in this work. This claim should be quantified and experimental data should be provided.

[Response 2]

In Refs. [41] and [42] as well as prior works on the streak-camera-based ultrafast imaging (e.g., Ref. [R1, R2]), the mainly employed reconstruction algorithms are single-view two-step iterative shrinkage/thresholding (TwIST) and dual-view TwIST. Compared to them, the main differences in SPLIT's reconstruction algorithm [i.e., the dual-view plug-and-play (PnP) alternating direction method of multipliers (ADMM) algorithm] are manifested in the algorithm's framework, the capability of the implemented state-of-the-art denoiser, and the amount of information in observation. To illustrate our point, we first provide a summary of the frameworks of TwIST and PnP-ADMM [see the ensuing sections (A) and (B)], followed by the description of the advantages of the latter [see the following section (C)]. To quantitatively demonstrate the superiority of SPLIT's reconstruction algorithm, we reconstructed the same experimental dataset using these three algorithms. These results are summarized in Supplementary Note 8 and Supplementary Fig. 6 of the revised manuscript.

(A) Framework of TwIST

TwIST implements a two-step version of iterative shrinkage/thresholding [R3], which is applied to solve an inverse problem (i.e., given y to find a solution to $y = Kx$) by a minimizer of a convex objective function:

$$f(x) = \operatorname{argmin}_x \left\{ \frac{1}{2} \|Kx - y\|^2 + \lambda \Phi(x) \right\}. \quad (\text{R1})$$

Here, x is the signal (e.g., video) to be reconstructed. K represents the operator applied to x arising from the physical forward model. $\Phi(\cdot)$ is the regularization function. λ is the known regularization constant. For the j^{th} iteration, the estimation of x becomes

$$x_1 = \mathcal{T}_\lambda(x_0), \quad (\text{R2})$$

$$x_{j+1} = (1 - \alpha)x_{j-1} + (\alpha - \beta)x_j + \beta \mathcal{T}_\lambda(x_j). \quad (\text{R3})$$

Here, x_0 is the vectorized signal. α and β are the pre-set constants for different reconstruction tasks, which affect the convergence rate of the minimization problem of Eq. (R1). The designation “two-step” stems from the fact that it depends on both x_{j-1} and x_j , rather than only on x_j . For $j \geq 1$, a function mapping operation $\mathcal{T}_\lambda: \mathbb{R}^m \rightarrow \mathbb{R}^m$ is defined as

$$\mathcal{T}_\lambda(x) = \Psi_\lambda[x + K^T(y - Kx)]. \quad (\text{R4})$$

Here, K^T denotes the transpose of K . Ψ_λ denotes a denoising operator, whose choice is related to the regularization function $\Phi(\cdot)$. With total variation (TV) as the regularization function, the denoising operator leverages the Chambolle algorithm [R4].

The relative change of the estimated output from the objective function [i.e., Eq. (R1)] is used as the merit function. The iteration process stops when this change is less than the pre-set tolerance value ϑ (e.g., 0.01), i.e.,

$$\frac{|f(x_j) - f(x_{j-1})|}{f(x_j)} \leq \vartheta. \quad (\text{R5})$$

The dual-view TwIST shares the same two-step iteration in the TwIST procedure but introduces another view to facilitate TwIST to retrieve better spatial features. Such a dual-view setting is reliable for computational imaging problems [R5] and has been used in recent works on streak-camera-based ultrafast optical imaging [R1, R6].

(B) Framework of PnP-ADMM

PnP-ADMM is based on the ADMM algorithm, which is an advanced tool for minimizing the sum of multiple separable functions. For simplicity, we use the two-function model in Eq. (R1) as an example. The algorithm works by converting the unconstrained optimization [i.e., Eq. (R1)] into a constrained problem by introducing a variable v :

$$(\hat{x}, \hat{v}) = \underset{x, v}{\operatorname{argmin}} \left\{ \frac{1}{2} \|Kx - y\|^2 + \lambda \Phi(x) \right\}, \text{ subject to } x = v. \quad (\text{R6})$$

It considers the augmented Lagrangian function by introducing a Lagrange multiplier u and penalty parameter ρ , so that Eq. (R6) becomes

$$(\hat{x}, \hat{v}, \hat{u}) = \underset{x, v, u}{\operatorname{argmin}} \{ \mathcal{L}(x, v, u) \}. \quad (\text{R7})$$

where $\mathcal{L}(x, v, u) = \frac{1}{2} \|Kx - y\|^2 + \lambda\Phi(v) + u^T(x - v) + \frac{\rho}{2} \|x - v\|^2$. Then, the algorithm finds the solution by seeking a saddle point of \mathcal{L} , which involves solving a sequence of sub-problems in the form

$$x^{(j+1)} = \operatorname{argmin}_{x \in \mathbb{R}^m} \left\{ \frac{1}{2} \|Kx - y\|^2 + \frac{\rho}{2} \|x - \tilde{x}^{(j)}\|^2 \right\}, \quad (\text{R8})$$

$$v^{(j+1)} = \operatorname{argmin}_{v \in \mathbb{R}^m} \left\{ \lambda\Phi(v) + \frac{\rho}{2} \|v - \tilde{v}^{(j)}\|^2 \right\}, \text{ and} \quad (\text{R9})$$

$$\bar{u}^{(j+1)} = \bar{u}^{(j)} + (x^{(j+1)} - v^{(j+1)}). \quad (\text{R10})$$

Here, $\bar{u}^{(j)} \stackrel{\text{def}}{=} u^{(j)}/\rho$ is the scaled Lagrange multiplier. $\tilde{x}^{(j)} \stackrel{\text{def}}{=} v^{(j)} - \bar{u}^{(j)}$ and $\tilde{v}^{(j)} \stackrel{\text{def}}{=} x^{(j+1)} + \bar{u}^{(j)}$ are the intermediate variables [R7]. Under the mild conditions, one can show that the iterates returned by Eqs. (R8)–(R10) converge to the solution of Eq. (R1).

The idea of PnP-ADMM is to modify Eq. (R9) by observing that it is a denoising step if we treat $\tilde{v}^{(j)}$ as a “noisy” version of v and $\Phi(v)$ as a regularization for v . Based on this observation, we can replace Eq. (R9) with a denoiser $\mathcal{D}_\sigma: \mathbb{R}^m \rightarrow \mathbb{R}^m$ such that

$$v^{(j+1)} = \mathcal{D}_\sigma(\tilde{v}^{(j)}), \quad (\text{R11})$$

where $\sigma = \sqrt{\lambda/\rho}$ is the denoising strength. The choice of \mathcal{D}_σ is broad, including TV denoising, deep convolution neural network, and the block-matching and 3D filtering (BM3D). PnP-ADMM uses the relative change of the estimate in adjacent iterations as the merit function:

$$\text{if } \frac{\|x^{(j+1)} - x^{(j)}\|_2}{\|x^{(j+1)}\|_2} < \varpi \text{ and } \rho^{j+1} = \rho^j. \quad (\text{R12})$$

Here, ϖ ($0 < \varpi < 10^{-3}$) is the pre-set tolerance value. By combining the dual-view strategy into PnP-ADMM, we constructed the dual-view PnP-ADMM SLPIT algorithm, which shares a similar iteration procedure with PnP-ADMM.

(C) Comparison between TwIST and PnP-ADMM

First, the PnP-ADMM framework [i.e., Eqs. (R6)–(R12)] has a better decomposability than that of the TwIST algorithm [i.e., Eqs. (R2)–(R5)] in handling complex and multiple-featured global optimization problems. Using a decomposition-coordination strategy, PnP-ADMM divides the large global optimization problem into small and easier-to-handle sub-problems, whose solutions are coordinated to help pinpoint the global minimization [R7]. For instance, the inverse problem

Eq. (R1) is separated into three sub-problems as Eqs. (R8)–(R10) in PnP-ADMM. Among them, Eq. (R9) is cast as a denoising step to leverage advanced denoising functions. More flexible than TwIST, PnP-ADMM is not restricted by a specific combination of regularizers and denoising operators but can choose many off-the-shelf denoising functions.

Second, the denoising function BM3D used in the PnP-ADMM algorithm has a better performance than the TV regularizer used in TwIST [R8]. TV exploits the gradient sparsity of a signal, which tends to introduce staircasing artifacts in the reconstruction. By contrast, as the referee has pointed out, BM3D is an advanced denoising method based on effective filtering in a 3D transform domain by combining the sliding-window transform process with block-matching [R9]. In the sliding process, blocks with similar spatial features and intensity levels are selected using the block-matching concept [R10]. These matched blocks are stacked to form a 3D array, and the data in the array exhibit high correlation. Then, a 3D de-correlating unitary transformation is applied to exploiting this correlation and effectively attenuating the noise by reducing the transform coefficients. Finally, using an inverse 3D transformation, all matched blocks are estimated. This procedure is repeated for each sliding window, and the final estimate is computed as a weighted average of all of those overlapping estimates.

Finally, akin to dual-view TwIST versus single-view TwIST, dual-view PnP-ADMM enriches the observation of the dynamic scene in data acquisition, which enables a better quality in reconstructed images (in terms of higher spatial resolution, higher image contrast, and fewer artifacts) than single-view PnP-ADMM. In particular, View 1 losslessly retains spatial information while discarding all temporal information. On the other hand, View 2 preserves temporal information due to the temporal shearing operation in data acquisition. As a result, these two views, as an optimal combination, enable one to record an optimum amount of information with the minimum number of measurements.

[Comment 3]

High resolution imaging is claimed in both the abstract and throughout the paper. This has to be put in the context of an imaged area, number of pixels in the imaging systems, optics, etc.

[Response 3]

We thank the referee for this suggestion. In the revised manuscript, whenever mentioned the resolution, we provided the information of the field of view.

[Comment 4]

The authors make a claim that this image sensor can be used for high accuracy thermal imaging and cite several papers where thermal imaging is critical. However, the paper provides convincing evidence that the imaging system can detect thermally modulated PLT. In other words, a UCNPs are necessary to sense the temperature of the underlying structure under the assumption that both entities have the same temperature. This is different when compared to how thermal cameras operate. These claims are misleading and should be corrected.

[Response 4]

We thank the referee for pointing out the difference between SPLIT and conventional temperature imaging using thermal cameras. First, we would like to clarify that we do not use “thermal imaging” anywhere in the original manuscript. We agree with the referee that SPLIT enables fast and accurate lifetime-based optical temperature sensing. In the revised manuscript, we have clarified that the herein devised imaging system by itself records only the lifetime images; yet, when using UCNPs as contrast agents, those images also carry temperature information *in situ*, where the UCNPs reside (please see Lines 269–271 on Pages 9–10).

In addition, the assumption that UCNPs have the same temperature as their environment holds true. UCNPs can quickly reach thermal equilibrium with their surrounding environment in powder or dispersion forms [R11-R13], which validates all the measured results in our work as well as numerous other reports of UCNP-based nanothermometry [R14-R17]. We have also verified this assertion by using a thermocouple and a thermal camera as references when performing temperature measurement experiments. This information has been described in Line 192 on Page 7 and can be seen in Supplementary Fig. 14b.

It is worth noting though, that we are aware that heat transfer at the nanoscale can break away from the Fourier law [R18], which can subsequently change thermalization dynamics between nanoparticles and surroundings. However, in the case of UCNPs, neither in single-particle nor ensemble studies, significant deviations between the UCNPs and the environment have been

observed so far, granted other temperature measurement artifacts were accounted for (e.g., excitation of upper energy states under high excitation power density irradiation [R13, R19]). Even then, these effects mainly influence the Boltzmann-driven population between the thermally coupled excited states of Er^{3+} , which are not used in our work. Finally, excitation power densities were also kept low to avoid these effects.

[Comment 5]

The key feature of this imaging technology is the PLT imaging capabilities with ~30 microseconds temporal resolution and leveraging sparsity in a scene to employ CS algorithms. This is the key strength of the work and achieved with an elegant approach. It would be good for this to be better explained rather than the focus to be on imaging temperature modulated PLT. The temperature modulated PLT can be part of the presented work but not necessarily the focus.

[Response 5]

We appreciate the referee's acknowledgment of the broad utility of our work. Indeed, SPLIT brings in ultrahigh imaging speeds to advanced cameras with a minimum addition in cost. In this regard, the ultrahigh-speed imaging system reported in this work could be used for many other novel applications. In the revised manuscript, we have added this discussion and more explanation in the last paragraph of Supplementary Note 12.

Despite being a generic platform, the ultrahigh-speed imaging system developed in this work is perfectly suitable for the fast 2D temperature mapping showcased in our manuscript. It brings single-shot wide-field photoluminescence lifetime imaging to UCNPs for the first time. The link of the lifetimes with the temperatures extends the measuring ability of this system to microscale thermometry as a new application of great value. Hence, SPLIT presents a striking example to show the usefulness of this imaging platform. For these reasons as well as to maintain a coherent description, we have decided to keep the main structure of our manuscript.

[Comment 6]

The time shearing effects should be better explained in the main text or methods. Since temporal and spatial resolution are tightly coupled in this imaging system, they should be elaborated in the text. It is hard to understand how was 33 kfps was achieved: the EMCCD operates at 16 fps and has 1024 by 1024 pixels. If the spatial resolution is 20 microns on a 1500 by 1500 micron sample, then the effective number of pixels are 75 by 75. This means temporal resolution can only be increased by a factor of 186 by time shearing or ~ 3 kfps will be the maximum frame rate. How was achieved higher temporal resolution? What is the effective spatial resolution?

[Response 6]

Time shearing mentioned in this Comment, which is called temporal shearing in our manuscript and is denoted by the operator \mathbf{S} , is described in the last paragraph on Page 4 of the original manuscript. In the revised manuscript, we have added a new paragraph in Section B of Methods and Supplementary Fig. 15 to further explain this operation.

We then explain the calculation of SPLIT's imaging speed. If we understand correctly, the referee estimated the maximum frame rate by multiplying the ratio of the camera's pixel counts to its effective pixel counts, i.e. $\left(\frac{1024}{75}\right)^2 = 186$, by the EMCCD camera's intrinsic frame rate (i.e., 16 fps with full pixel count), which gives $186 \times 16 \text{ fps} = \sim 3 \text{ kfps}$. However, this way of calculation does not reflect how the SPLIT system is operated in our work. First, according to Eq. (M1) in Methods, SPLIT's imaging speed is controlled by the galvanometer scanner (GS)'s rotation, Lens 5's focal length, and the EMCCD's pixel size, but *not* by the EMCCD's intrinsic frame rate. In addition, the exposure time of the EMCCD camera is set by the users according to the sequence depth N_t and the targeted imaging speed, *not* by the reciprocal of its intrinsic frame rate. Thus, in our work, although the EMCCD camera ran at 20 Hz, the exposure time was set to $t_e = 3 \text{ ms}$, much shorter than the reciprocal of the frame rate (i.e., $1/20 \text{ Hz} = 50 \text{ ms}$). Under these settings, the SPLIT system compressively records up to 100 frames in 3 ms, which hence has an imaging speed of up to 33 kfps.

SPLIT's spatial resolution mainly depends on the objective lens and the dynamic scene. In this work, the objective lens ($4\times$ magnification ratio, 0.1 numerical aperture, and 11 mm field number) determines the optics-limited spatial resolution to be $\sim 4 \mu\text{m}$. Meanwhile, the sparsity of the dynamic scene and the signal-to-noise ratios (SNRs) of the acquired snapshot exert an impact on the selection of the encoding pixel size. Larger encoding pixel size facilitates the image

reconstruction, especially under the low-SNR scenario. However, it also reduces the spatial and temporal resolutions. In our study, we found encoding pixels with a $60\ \mu\text{m} \times 60\ \mu\text{m}$ size produced the best balance. Under these conditions, in Fig. 2e, even though the spatial features of these bar patterns remain sharp, we conservatively determine the spatial resolution to be $20\ \mu\text{m}$ by using the reconstructed intensity as the criterion. Thus, the effective spatial resolution is $20\ \mu\text{m}$, and we agree with the referee that the effective number of pixels is 75×75 in a field of view (FOV) of $1.5\ \text{mm} \times 1.5\ \text{mm}$. Currently, this number is limited by the employed objective lens, the spatial profile of the laser, and the characteristics of the UCNPs used in our work. The 11-mm field number of this objective lens limits the maximum FOV to 2.75 mm in diameter. The actual FOV needs to be smaller to avoid image distortion and vignetting at the edge. Moreover, the efficiency of UCNPs is nonlinear to the excitation laser power density. Because the excitation laser pulse has a Gaussian spatial profile, the UCNPs at the periphery of the FOV do not have a strong emission. Meanwhile, however, an excessive excitation power density could induce intensity-dependent variation in lifetimes, which we must avoid in our experiments (details are provided in Response 4 to Referee 2). Thus, to balance the SNRs and fidelity in SPLIT's measurement, especially for the imaging experiments underneath beef and chicken tissues, we had to sacrifice some FOV.

Finally, the technical specifications of SPLIT could be further improved according to the requirements of targeted applications. For example, the imaging speed can be easily increased by applying a higher voltage to the GS. The spatial resolution could also be enhanced by using an objective lens with a higher numerical aperture and magnification ratio. The FOV could be expanded by using an objective lens with a larger field number and a beam homogenizer to generate a flattop profile.

[Comment 7]

The experiments of imaging UCNP under different chicken phantoms should be improved. A better scattering medium should be chosen since the chosen phantoms don't have water or blood which are the main scattering culprits.

[Response 7]

We thank the referee for this suggestion. We would first like to clarify that the phantom experiment shown in the original manuscript was conducted by using fresh chicken breast tissue, which had

water in it. We have added this information to the revised manuscript. Second, we conducted another phantom experiment using fresh beef tissue (purchased from a supermarket) as a scattering medium, which contains both water and myoglobin. Both the oxygenated myoglobin and the deoxygenated myoglobin have similar optical absorption properties to those of the oxygenated and the deoxygenated hemoglobin [R20]. They are also major chromophores in muscle [R21]. In addition, due to the COVID-19, it is extremely time-consuming and difficult to obtain an approved new animal protocol to acquire fresh animal tissue, especially to INRS-EMT that does not have an animal facility in its proximity. Therefore, we used the beef tissue as the substitute to mimic the scattering medium in which both water and blood were present. These results successfully demonstrated SPLIT's imaging capability when the chosen phantom has both absorption and scattering. For details, please check the third paragraph in Supplementary Note 10 and Supplementary Fig. 10 in the revised manuscript.

[Comment 8]

The proposed imaging system should be more carefully evaluated against high-speed commercial cameras (ref. 36, 37 and other cameras with fps of ~100K). Also if the argument about temperature sensing in PLT is maintained in the paper, then a comparison with FLIR camera should be provided in the paper (spatial and temporal resolution, sensitivity, size, power, cost, etc).

[Response 8]

First, as we have explained in the second paragraph on Page 3, the cameras in Refs. [36] and [37] of the original manuscript (which are Refs. [37] and [38] in the revised manuscript) work by combining an image intensifier with the wide-field time-correlated single-photon counting (TCSPC) technique. Hence, these cameras require numerous repetitive measurements to obtain a single 2D lifetime map, which leads to a long data acquisition time (10,000 seconds in Ref. [37] and 60 seconds in Ref. [38]). This approach thus also requires the photoluminescence emission to be precisely repeatable. By contrast, SPLIT finishes data acquisition in a single exposure of 3 ms, which not only drastically improves the efficiency of data acquisition but also enables it to measure non-repeatable lifetime distributions (e.g., a moving photoluminescent sample shown in Fig. 4) in a wide field.

Second, most cameras with frame rates of >20 kfps need specially designed sensors (e.g., the in-situ image storage CCD sensor reported in Ref. [R22]), which usually have limited filling factors and hence are not suitable for low-light applications like photoluminescence lifetime imaging. In addition, their prices are considerably higher than that of the SPLIT system. Four commercial high-speed cameras are listed in Table R1 with their prices (excluding taxes; as of June 23, 2021), frame rates, and pixel counts. SPLIT clearly shows advantages in its technical specifications with its cost-efficiency. We would also like to emphasize that the majority of the cost of SPLIT is in the EMCCD and CMOS cameras, and less than US\$5,000 is needed to purchase the optics and electronics to enable ultrahigh imaging speeds on the EMCCD camera. Considering the wide availability of these cameras in academia and industry as well as the wide applicability of SPLIT's imaging concept to a wide range of cameras, SPLIT is the advantageous and economical strategy that can bring ultrahigh imaging speeds to numerous types of cameras for specific studies.

Table R1 Comparison between commercial high-speed cameras and SPLIT

Part number	Price (USD)	Imaging speed (fps)	Pixel count	Manufacturer
FASTCAM SA-Z	~150,000	20,000	1024×1024	Photron
TMX-6410	~180,000	65,940	1280×800	PHANTOM
i-SPEED 727	~134,000	50,000	840×606	iX-Cameras
HPV-X2	~250,000	10,000,000	400×250	Shimadzu
SPLIT	~65,000 ^[Note 1]	33,000 ^[Note 2]	460×460 ^[Note 3]	INRS

[Note 1]: The cost of SPLIT consists of two portions: (1) ~US\$60,000 for the EMCCD camera and the CMOS camera and (2) ~US\$5,000 for optics and electronics.

[Note 2]: By applying a higher voltage to the GS, the imaging speed of the SPLIT system is tunable to several million fps.

[Note 3]: By using an objective lens with a larger field number and greater excitation power density, the pixel count of the SPLIT system can be increased to 1024×1024.

Finally, we experimentally compared the image quality between SPLIT and a thermal camera (Yoseen X384D). This camera was chosen based on the instrument's availability as well as its similar pixel count to that of the FLIR E8 thermal camera. The results show superior image

quality of SPLIT to thermal imaging for temperature mapping. In particular, images produced by SPLIT have much higher contrast. Meanwhile, SPLIT is more resilient to interference from environmental radiation. The experimental details and results have been included in Supplementary Note 13 and Supplementary Fig.14 in the revised manuscript.

REFeree #2

[Comment 0]

Liu et al. reported a time-domain wide-field lifetime imaging system. Combined with Yb³⁺/Er³⁺ nanothermometers, this system realized the intracellular temperature mapping in onion cells. This is an interesting work, but this reviewer feels that the originality and experimental findings did not meet the standards of Nature Communications.

[Response 0]

We thank the referee for acknowledging that our work is interesting. To facilitate the referee for a better assessment, we summarize the originality and experimental findings of our work here:

As a one-of-its-kind modality in photoluminescence temperature sensing, SPLIT possesses the following attractive advantages compared to existing instrumentations.

- (1) From the standpoint of system development, a new dual-view hardware design maximally preserves both spatial and temporal information, and a new reconstruction algorithm reconstructs videos with premier quality.
- (2) SPLIT can be coupled to a variety of UCNP temperature indicators with a wide span of photoluminescence lifetimes, which overcomes the limitation in existing frequency-domain photoluminescence lifetime imaging techniques.
- (3) SPLIT generates a temperature map using a single exposure, which largely improves the imaging speed and accuracy compared to existing scanning-based techniques.
- (4) To our knowledge, SPLIT is the only time-domain modality with video-rate lifetime/temperature mapping ability.
- (5) The single-shot wide-field data acquisition in SPLIT extends the photoluminescence lifetime imaging to the observation of non-repeatable temperature dynamics for the first time.

Besides pushing forward the technical frontier, SPLIT has great potential in many fields of studies, including materials science and biomedicine.

- (1) The near-infrared illumination facilitates SPLIT in longitudinal wide-field temperature mapping beneath the surface. This feature makes SPLIT a potent candidate for diverse applications from temperature-regulated photothermal therapy of melanoma to temperature-indicated stress analysis in turbines.

(2) SPLIT has accomplished the first-ever wide-field temperature tracking of a moving biological sample at single-cell resolution, which will open up many opportunities in studying temperature-regulated cellular activities (e.g., apoptosis and mitochondria activities). The high-resolution dynamic imaging ability could also enable temperature monitoring in chemical reactions and physical processes using UCNP labeling.

[Comment 1]

The mechanical streaking-based lifetime imaging introduced in this work was reported previously with the same principle (Opt. Express 2020, 28, 26717-26723). Instead of using a mask, the previous reported work used a line-scanning confocal set-up.

[Response 1]

We would like to emphasize that SPLIT is significantly different from the line-scanning confocal setup (reported in the reference mentioned by the referee, which is also cited as Supplementary Ref. [25] in the revised manuscript) in terms of working principle, application scope, and experimental findings. In particular, that work, along with many other previous works (e.g. [R23-R25]), belongs to scanning-based photoluminescence lifetime measurements using streak cameras. However, these works have a severe limitation. The line-scanning method can only get an (y, t) slice at a particular x position, because the conventional operation of the streak camera does not allow mixing the temporal information with the spatial information in the x axis. This limitation leads to the requirement of repetitive measurements. Thus, these works fall short when the 2D photoluminescent events cannot be repeated. Please see an experimental demonstration in Supplementary Fig. 13 and the associated explanation in Supplementary Note 12.

Herein presented SPLIT system uses a novel compressed-sensing-aided streak imaging paradigm to overcome this limitation. The 2D encoding mask and the ensuing temporal shearing operation attach independent prior information to each frame in a 2D transient scene of photoluminescence. This mask thus enables retaining the (x, y) information at each time point. Meanwhile, the prior information allows for an intermix between the spatial and temporal information in the temporal shearing direction. Leveraging a new reconstruction algorithm, SPLIT retrieves an (x, y, t) datacube of the wide-field photoluminescence event in a single shot. Finally, we applied this technique with advanced photoluminescent materials—UNCPs. Using their

thermally dependent energy exchange dynamics between Er^{3+} excited states, we successfully convert wide-field lifetime sensing to dynamic temperature mapping. These important experimental findings open up a new application of ultrahigh-speed optical imaging and advance the frontier of photoluminescence lifetime imaging. It also expands the scope of UCNP-lifetime-based temperature sensing to moving samples for the first time (demonstrated in Fig. 4). Thus, SPLIT largely exceeds the multiple-shot line-scanning confocal setups in imaging capability and hence will find broader applications. More information and detailed explanations of the aforementioned summary are provided in Supplementary Note 12 and Supplementary Table 1.

[Comment 2]

The thermal responsive mechanism of $\text{Yb}^{3+}/\text{Er}^{3+}$ is not studied in this work, however, similar phenomena was reported before (Nature Photon. 2018, 12, 154–158).

[Response 2]

We would like to clarify that our work *by no means explores or exploits* the mechanism of temperature-dependent intensity variation in the upconversion emission reported in the reference provided by the referee. This subject is completely separate from our study. In fact, the emission thermal enhancement phenomenon cannot be observed with our UCNPs because (1) our UCNPs feature thick passive outer shell, that prevents interaction between the lanthanide dopants in the core and the environment, and (2) our experiments were carried out at the physiologic temperatures (293–319 K), well below the 400–450 K temperature range where the referenced phenomenon occurs.

In our work, we turned our attention to relatively little explored lifetime-based temperature sensing with UCNPs. As we have pointed out in Lines 49–63 on Pages 2–3 of the manuscript, the advantage of using lifetime-based thermometry is that, unlike intensity-based temperature readout, it is insensitive to artifacts such as instrument spectral response, tissue absorption, and scattering.

The mechanism by which the lifetime of a particular excited state of Er^{3+} changes with temperature, is largely ascribed to the temperature-dependence of multiphonon relaxation between the neighboring energy levels, as we had described in the manuscript (please see Lines 201–206 on Page 7). Thus, our work does not focus on the explicit investigation of mechanism(s) that govern the thermal-dependence of upconversion emission intensity or excited states lifetimes, but

rather on how to practically employ the latter phenomenon for lifetime-based temperature mapping with easily accessible, accurate, and powerful instrumentation, such as the SPLIT technique.

[Comment 3]

The thermal uncertainty (or thermal resolution) by lifetime measurement is not given in this context.

[Response 3]

In the original manuscript, we have provided the thermal sensitivity of our UCNPs [see Eq. (S14)], which is linked to the thermal uncertainty. In the revised manuscript, to enhance the completeness of our work, we have calculated the thermal uncertainty. Please see this new content in the third paragraph in Supplementary Note 9 in the revised manuscript.

[Comment 4]

The upconversion process is power dependent, and the lifetime of the green or red emission band of Er via upconversion changes in different excitation power density. The established lifetime-temperature values can not be applied in different power density (like through the chicken breast).

[Response 4]

Although the referee is correct about the excitation power density dependence for the lifetime of different Er³⁺ excited states, such changes occur only under specific conditions—in heavily Er³⁺-doped UCNPs (>20 mol%) and under very high excitation power density (e.g., >100 W/cm²). Under mild excitation power densities, their lifetimes do not vary [R26]. For the experiments in our work, neither condition was satisfied—UCNPs (with only 2 mol% Er³⁺) were excited in the range of 6–40 W/cm² at the sample surface. For the phantom experiments with the chicken and beef tissues, the intensity only decreases with the increase in depth. Therefore, there should be no power density-dependent fluctuation of excited states lifetimes in our UCNPs under present experimental conditions.

In fact, the invariance of lifetimes to the excitation power density has been directly confirmed from the experiments described in Supplementary Note 10 and Supplementary Fig. 10g

in the revised manuscript, which show that the measured lifetime values do not change with different thicknesses of beef tissue, hence independent to the excitation intensity.

Finally, the lifetimes of UCNPs have been previously used to acquire functional information subcutaneously [R27-R29]. In these works, the lifetime-temperature relation (or lifetime-multiplexing) was established during the stage of material characterization and then applied to functional imaging with biological tissue. SPLIT follows the same protocol. In this regard, the successful implementation of UCNP lifetimes in tissue imaging provides additional affirmation that the established lifetime-temperature relations in our work can be applied to temperature mapping in different power densities.

[Comment 5]

Although this work demonstrated the experiments with chicken breast, as widely acknowledged, the emission wavelength in the visible region is not suitable for deep tissue imaging.

[Response 5]

We would like to clarify that the investigation of deep tissue imaging is *not* by any means the research focus of our manuscript. Rather, from the standpoint of novel applications, the major significance of SPLIT lies in the first-ever single-shot wide-field photoluminescence lifetime imaging using UCNPs and its application to dynamic temperature mapping. Meanwhile, the chicken tissue phantom experiments aim to demonstrate the feasibility of the SPLIT system in the biological environment (please see Line 208 on Page 8). We agree with the referee that visible light has difficulties penetrating deep tissue. However, it can still be used in numerous biomedical experiments. As mentioned in the last paragraph of Discussion in the manuscript, of particular relevance to our work is SPLIT's potential application in temperature-based early diagnostics and temperature-regulated photothermal therapy of micro-melanoma [R30, R31]. In these cases, an imaging depth of several tens to hundreds of micrometers is required, which has been demonstrated in our experimental results (please see Supplementary Note 10 and Supplementary Figs. 8–10).

We also would like to point out that SPLIT, even with its existing configurations, has great potential for deep tissue imaging. First, we have experimentally demonstrated SPLIT using the red emission of the core/shell $\text{NaGdF}_4:\text{Er}^{3+}, \text{Yb}^{3+}/\text{NaGdF}_4$ UCNPs with a center wavelength of 660 nm. Using chicken breast tissue as a scattering medium, we demonstrated a penetration depth of

0.75 mm. Since this wavelength approaches the first biological window, it could be used for tissue imaging at millimeter-level depth, as many existing works have explored [R32-R34]. Second, as a generic imaging platform, SPLIT can be easily adapted to deep tissue imaging. As a relatively straightforward step, we could build the system with near-infrared optics and short-wavelength infrared cameras for imaging the Stokes emission of lanthanide-doped nanoparticles. This statement is included in the penultimate sentence of the last paragraph in the Discussion of the manuscript.

REFEREE #3

[Comment 0]

The authors have developed a new luminescence lifetime imaging technique that combines controlled illumination and smart image acquisition through a coded aperture to enable compressed sampling of image data in space and time at extreme temporal resolution.

[Response 0]

We thank the referee for acknowledging the innovations in our work.

[Comment 1]

The authors should compare their work to this particular prior publication: Lixin Liu, Yahui Li, Luogeng Sun, Heng Li, Xiao Peng, Junle Qu, "Fluorescence lifetime imaging microscopy using a streak camera," Proc. SPIE 8948, Multiphoton Microscopy in the Biomedical Sciences XIV, 89482L (28 February 2014); <https://doi.org/10.1117/12.2039056>, who used galvomirrors only to attempt to obtain similar results, this may help justify the benefits of an indirect imaging method using scanning and a coded aperture.

[Response 1]

We thank the referee for pointing out this reference, which we have added in the revised manuscript. Compared to this reference, the differences and the benefits of SPLIT are mainly manifested in three aspects.

First, the GS was used for different functions. In this reference, the GS was used at the illumination side to steer the incident laser beam for line scanning. In contrast, in SPLIT, the GS is used at the detection side as an ultrahigh-speed sweeping unit for the temporal shearing operation.

Second, the application scope of the referenced work is narrower than SPLIT. Due to its line scanning operation, the system presented in the reference requires many measurements to form one 2D fluorescence lifetime map, which thus requires the photoluminescence events to be precisely repeatable. In contrast, SPLIT can generate a 2D lifetime map in a single acquisition, which not only brings in high throughput and a high work efficiency but also makes it perfectly suitable for sensing non-repeatable or difficult-to-reproduce events, such as a moving photoluminescent sample shown in Fig. 4 in Main Text.

Finally, the technical specifications and the targeted applications are different. The referenced work was employed for fluorescence lifetime imaging (at the picosecond-to-nanosecond level), while SPLIT is used for photoluminescence lifetime imaging of UCNP (at the microsecond-to-millisecond level). Considering the largely different time scales of these two types of processes, each imaging system is optimized for its targeted application. Compared to the reference, SPLIT has a lower frame rate, but considerably higher quantum efficiency and higher image quality. Thus, SPLIT is well-positioned for dynamic photoluminescence lifetime mapping in 2D.

The details of this comparison have been shown in Supplementary Note 12 and Supplementary Table 1.

[Comment 2]

The authors should also compare their work to ref. 48, in greater details, since this imaging method also used a coded aperture to intermix spatial and temporal information, perhaps taking away some of the novelty claims of the paper. Is the novelty limited to the denoising algorithm? If so, please clearly indicate if the setup in figure 1 has been previously used by others, and if there are minor variations in the design and operation, please explain what those specific contributions are and how this experimental setup differs from techniques previously published.

[Response 2]

We have provided a comparison between SPLIT and CUSP (reported in Ref. [48]) in Supplementary Note 12 and Supplementary Table 1 in the revised manuscript.

The SPLIT system has innovations in hardware design in the following four aspects. First, SPLIT's setup, shown in Fig. 1 in Main Text, is new and has never been previously used by others. The dual-view hardware design is implemented with an ultrahigh-speed mechanical streak camera for the first time.

Second, compared to this reference, the SPLIT system uses a transmissive mask, instead of a digital micromirror device (DMD), to considerably improve the efficiency in spatial encoding and imaging quality. When a binary encoding pattern is loaded onto the DMD, each micromirror is tilted to $+12^\circ$ or -12° as the "1" or "0" pixels, which makes the DMD a reflective blazed grating. When the incident light is not at the blaze wavelength, the diffraction efficiency can be rather

limited. The overall light modulation efficiency is further reduced by the reflection from the DMD's cover glass and by its limited fill factor. In addition, for CUSP, the DMD is required to be placed in the Littrow configuration to retro-reflect the incident light. Blurring the encoding pattern at the periphery, this design thus restricts CUSP's FOV even if an objective lens with a large field number is used [R35]. In contrast, in the SPLIT system, the light has normal incidence to the transmissive mask. This operation guarantees uniform light modulation efficiency to different wavelengths and lifts the restriction of the FOV. In addition, there is no gap between adjacent pixels (i.e., a 100% fill factor). Without a cover glass, the light does not get attenuated due to multiple reflections. Both features reduce the light loss in spatial encoding.

Third, SPLIT uses an EMCCD camera with a GS, instead of an optoelectronic streak camera, for time-resolved imaging. The EMCCD camera has a much higher quantum yield (>90%) than that of an optoelectronic streak camera (<15%), which provides superior SNRs in the captured snapshot and thus better image quality in reconstruction. The all-optical operation also avoids the space-charge effect in the optoelectronic streak camera, which improves spatial resolution and dynamic range (e.g., >60,000 of the EMCCD camera used in this work). These advantages endow SPLIT with superior image capability.

Finally, the CUSP technology is incapable of accomplishing the work presented in our manuscript. The longest sweeping time of the streak camera used in CUSP (i.e., Hamamatsu C6138) is 1 ns [R36]. Thus, this technique cannot be applied to sensing the microsecond-level lifetimes. In addition, CUSP used a 532-nm picosecond laser source, unsuitable to excite UCNPs. The high instantaneous intensity of this picosecond laser could also pose a higher risk for sample damage.

Besides the hardware innovation, as the referee pointed out, the reconstruction algorithm is indeed a novel aspect of SPLIT compared to the CUSP technique. The dual-view PnP-ADMM software leverages the state-of-the-art denoising algorithm to largely improve the reconstructed image quality. Please see the detailed explanation in our Response 2 to Referee 1. To better illustrate this point, we have added a comparison of qualities of images reconstructed by the dual-view PnP-ADMM, the dual-view TwIST algorithm (used in the CUSP technique), and the single-view TwIST algorithm (used in many other single-shot compressed temporal imaging modalities) in Supplementary Note 8 and Supplementary Fig. 6 of the revised manuscript.

[Comment 3]

Besides those two minor comments, the experiments validate the claims of performance quite clearly, the imaging capabilities are exciting and valuable to a broad community, and this reviewer, therefore, believes that the article is suitable for publication in this journal.

[Response 3]

We sincerely appreciate the referee for acknowledging the quality of our manuscript and work, and for recommending the publication of this paper.

REFERENCES

- R1. Liang J., Zhu L., and Wang L.V., *Single-shot real-time femtosecond imaging of temporal focusing*. Light: Science & Applications, 2018. **7**: 42.
- R2. Wang P., Liang J., and Wang L.V., *Single-shot ultrafast imaging attaining 70 trillion frames per second*. Nature Communications, 2020. **11**: 2091.
- R3. Bioucas-Dias J.M. and Figueiredo M. A., *A new TwIST: Two-step iterative shrinkage/thresholding algorithms for image restoration*. IEEE Transactions on Image processing, 2007. **16**(12): p. 2992-3004.
- R4. Chambolle, A. *An algorithm for total variation minimization and applications*. Journal of Mathematical imaging and vision, 2004. **20**(1): p. 89-97.
- R5. Shrestha S., et al., *Computational imaging with multi-camera time-of-flight systems*. ACM Transactions on Graphics (ToG), 2016. **35**(4): p. 1-11.
- R6. Liang J., et al., *Single-shot real-time video recording of a photonic Mach cone induced by a scattered light pulse*. Science advances, 2017. **3**(1): e1601814.
- R7. Boyd S., Parikh N., and Chu E., *Distributed optimization and statistical learning via the alternating direction method of multipliers*. 2011: Now Publishers Inc.
- R8. Lai Y., et al., *Single - Shot Ultraviolet Compressed Ultrafast Photography*. Laser & Photonics Reviews, 2020. **14**(10): 2000122.
- R9. Danielyan A., Katkovnik V., and Egiazarian K., *BM3D frames and variational image deblurring*. IEEE Transactions on Image Processing, 2011. **21**(4): p. 1715-1728.
- R10. Dabov K., et al., *Image denoising by sparse 3-D transform-domain collaborative filtering*. IEEE Transactions on image processing, 2007. **16**(8): p. 2080-2095.
- R11. Bastos A.R., et al., *Thermal properties of lipid bilayers derived from the transient heating regime of upconverting nanoparticles*. Nanoscale, 2020. **12**(47): p. 24169-24176.
- R12. Skripka A., et al., *Advancing neodymium single-band nanothermometry*. Nanoscale, 2019. **11**(23): p. 11322-11330.
- R13. Pickel A.D., et al., *Apparent self-heating of individual upconverting nanoparticle thermometers*. Nature Communications, 2018. **9**: 4907.
- R14. Savchuk O.A., et al., *Er: Yb: NaY 2 F 5 O up-converting nanoparticles for sub-tissue fluorescence lifetime thermal sensing*. Nanoscale, 2014. **6**(16): p. 9727-9733.
- R15. Brites C.D., et al., *Instantaneous ballistic velocity of suspended Brownian nanocrystals measured by upconversion nanothermometry*. Nature Nanotechnology, 2016. **11**(10): p. 851-856.
- R16. Zhu X., et al., *Temperature-feedback upconversion nanocomposite for accurate photothermal therapy at facile temperature*. Nature Communications, 2016. **7**: 10437.
- R17. Hartman T., et al., *Operando monitoring of temperature and active species at the single catalyst particle level*. Nature Catalysis, 2019. **2**(11): p. 986-996.
- R18. Cunha J., et al., *Controlling Light, Heat, and Vibrations in Plasmonics and Phononics*. Advanced Optical Materials, 2020. **8**(24): 2001225.
- R19. Martins J.C., et al., *Primary Luminescent Nanothermometers for Temperature Measurements Reliability Assessment*. Advanced Photonics Research, 2021: 2000169.
- R20. Arakaki L.S., Burns D.H., and Kushmerick M.J., *Accurate myoglobin oxygen saturation by optical spectroscopy measured in blood-perfused rat muscle*. Applied spectroscopy, 2007. **61**(9): p. 978-985.
- R21. Lin L., et al., *In vivo photoacoustic tomography of myoglobin oxygen saturation*. Journal of biomedical optics, 2015. **21**(6): p. 061002.

- R22. Etoh T.G., et al., *Evolution of ultra-high-speed CCD imagers*. Plasma and Fusion Research, 2007. **2**: p. S1021-S1021.
- R23. Liu X., et al., *Fast fluorescence lifetime imaging techniques: A review on challenge and development*. Journal of Innovative Optical Health Sciences, 2019. **12**(05): 1930003.
- R24. Biskup C., Zimmer T., and Benndorf K., *FRET between cardiac Na⁺ channel subunits measured with a confocal microscope and a streak camera*. Nature Biotechnology, 2004. **22**(2): p. 220-224.
- R25. Krishnan R., et al., *Development of a multiphoton fluorescence lifetime imaging microscopy system using a streak camera*. Review of Scientific Instruments, 2003. **74**(5): p. 2714-2721.
- R26. Teitelboim A., et al., *Energy transfer networks within upconverting nanoparticles are complex systems with collective, robust, and history-dependent dynamics*. The Journal of Physical Chemistry C, 2019. **123**(4): p. 2678-2689.
- R27. Li H., et al., *Temporal multiplexed in vivo upconversion imaging*. Journal of the American Chemical Society, 2020. **142**(4): p. 2023-2030.
- R28. Savchuk O. A., et al., *Er:Yb:NaY₂F₅O up-converting nanoparticles for sub-tissue fluorescence lifetime thermal sensing*. Nanoscale, 2014. **6**(16): p. 9727-9733.
- R29. Qiu X., et al., *Ratiometric upconversion nanothermometry with dual emission at the same wavelength decoded via a time-resolved technique*. Nature communications, 2020. **11**(1): p. 1-9.
- R30. Paul P. S. *Micromelanomas: a review of melanomas \leq 2 mm and a case report*. Case Reports in Oncological Medicine, 2014. **2014**: p. 206260.
- R31. Megaris A., et al., *Dermoscopy features of melanomas with a diameter up to 5 mm (micromelanomas): A retrospective study*. Journal of American Academy of Dermatology 2020. **83**(4): p. 1160-1161.
- R32. Kuchimaru T., et al., *A luciferin analogue generating near-infrared bioluminescence achieves highly sensitive deep-tissue imaging*. Nature Communications, 2016. **7**: 11856.
- R33. Matthews T.E., et al., *Deep tissue imaging using spectroscopic analysis of multiply scattered light*. Optica, 2014. **1**(2): p. 105-111.
- R34. Zambito G., et al., *Red-shifted click beetle luciferase mutant expands the multicolor bioluminescent palette for deep tissue imaging*. Iscience, 2021. **24**(1): p. 101986.
- R35. Lai Y, et al., *Single-shot ultraviolet compressed ultrafast photography*. Laser & Photonics Reviews, 2020. **14**(10): p. 2000122.
- R36. Femtosecond Streak Camera C6138 (FESCA-200), HAMAMATSU, URL: http://hamamatsu.com.cn/UserFiles/DownFile/Product/e_c6138.pdf. Accessed on 2021-07-02.

REVIEWERS' COMMENTS

Reviewer #1 (Remarks to the Author):

The authors have significantly improved the quality of the manuscript and addressed all comments and concerns raised by this reviewer.

Reviewer #2 (Remarks to the Author):

The authors have revised their manuscript well. I agree that the main contribution of this work is the development of imaging technique. Thus the current version is suitable to be published. Maybe further work can be done with a more suitable lifetime-based sensing system to show the superiorities of this system better in another manuscript.

RESPONSES TO REFEREES

We thank the referees for their insightful comments, which have helped us improve the quality of our manuscript.

REFEREE #1:

[Comment]

The authors have significantly improved the quality of the manuscript and addressed all comments and concerns raised by this reviewer.

[Response]

We thank the referee for acknowledging that we have addressed all the comments and concerns.

REFEREE #2:

[Comment]

The authors have revised their manuscript well. I agree that the main contribution of this work is the development of imaging technique. Thus the current version is suitable to be published. Maybe further work can be done with a more suitable lifetime-based sensing system to show the superiorities of this system better in another manuscript.

[Response]

We thank the referee for acknowledging that we have well revised the manuscript. We also thank the referee for acknowledging our contribution to the development of this imaging technique and recommending the publication of this manuscript. Finally, we appreciate the referee's suggestion for future research.